# Discovery of tumoricidal DNA oligonucleotides by response-directed in vitro evolution

Noam Mamet[1,2,6], Yaniv Amir[3,6]*, Erez Lavi[3,6], Liron Bassali[1], Gil Harari[1], Itai Rusinek[1], Nir Skalka[3], Elinor Debby[3], Mor Greenberg[1], Adva Zamir[1], Anastasia Paz[3], Neria Reiss[3], Gil Loewenthal[1], Irit Avivi[4], Avichai Shimoni[5], Guy Neev[3], Almogit Abu-Horowitz[1] & Ido Bachelet [1]*

Drug discovery is challenged by ineffectiveness of drugs against variable and evolving diseases, and adverse effects due to poor selectivity. We describe a robust platform which potentially addresses these limitations. The platform enables rapid discovery of DNA oligonucleotides evolved in vitro for exerting specific and selective biological responses in target cells. The process operates without a priori target knowledge (mutations, biomarkers, etc). We report the discovery of oligonucleotides with direct, selective cytotoxicity towards cell lines, as well as patient-derived solid and hematological tumors. A specific oligonucleotide termed E8, induced selective apoptosis in triple-negative breast cancer (TNBC) cells. Polyethylene glycol-modified E8 exhibited favorable biodistribution in animals, persisting in tumors up to 48-hours after injection. E8 inhibited tumors by 50% within 10 days of treatment in patient-derived xenograft mice, and was effective in ex vivo organ cultures from chemotherapy-resistant TNBC patients. These findings highlight a drug discovery model which is target-tailored and on-demand.

[1] Augmanity, Rehovot, Israel. [2] Faculty of Life Sciences, Bar-Ilan University, Ramat-Gan, Israel. [3] Aummune, Tel Aviv Sourasky Medical Center, Tel Aviv, Israel. [4] Tel Aviv Sourasky Medical Center, Sackler Faculty of Medicine, Tel Aviv University, Tel Aviv, Israel. [5] BMT Department, Division of Hematology, Sheba Medical Center Tel Hashomer, Ramat-Gan, Israel. [6] These authors contributed equally: Noam Mamet, Yaniv Amir, Erez Lavi. *email: yaniv@aummune.tech; ido@augm.com

Effect and selectivity are essential requirements for therapeutic molecules. However, it has become increasingly clear that for many severe diseases, achieving these requirements could be challenging. The continual emergence of drug resistance in cancer, for example, makes therapeutic targeting extremely difficult[1–3]. The problem is compounded by the high variability and patient heterogeneity of the disease[4], making it challenging for a single drug or protocol to be both effective and safe across many patients[5]. New drugs continue to be developed despite known resistance to them and the prediction that they will be effective only for a small fraction of patients[5,6]. The current premise of personalized medicine typically refers to predicting or validating responses to drugs from the set of currently available ones[7], leaving the problems of emergent resistance and off-target toxicity of these drugs unaddressed. Although superior to older generation chemotherapy in many ways, antibodies are specific to their antigens and would show selectivity only when antigen expression is limited to a specific target cell. Recently approved chimeric antigen receptor (CAR)-T cell therapies, while new and promising, have often shown adverse effects due to this fact[8].

For the purposes of this study we use cancer as a case study, and argue that an effective and viable therapeutic strategy for this disease would have to satisfy three requirements: (1) It needs to be tailored to a specific tumor/patient, due to the observed variability between individual cases; (2) It needs to be selective, to minimize adverse effects or eliminate them completely; and (3) Its discovery needs to be rapid and economically repeatable, to counter the emergence of resistance.

In this article, we describe a platform that essentially fulfills these requirements. While further development and improvement are necessary to expand it and establish its clinical potential in cancer and other conditions, we report extremely promising results that should motivate this effort. This platform is based on the in vitro evolution of oligonucleotides driven directly by a therapeutic response.

The ability to artificially evolve and select nucleic acid molecules with specific properties has been known for nearly three decades[9,10] and has produced diverse functions[11–15], molecular and cellular specificities[16–18], and therapeutic effects[19–21]. The SELEX[15,16] (systematic evolution of ligands by exponential enrichment) method is routinely used to find aptamers - RNA or DNA oligonucleotides with the ability to bind a specific molecular or cellular target. In SELEX, iterating rounds of selection are applied to an initial population of $10^9$–$10^{15}$ oligonucleotides. Selection pressure drives this population towards a subpopulation enriched with oligonucleotides capable of binding the target presented to them. The process is designed such that the oligonucleotides that are best binders survive each round and are passed on to the next one. Stringent washing steps are applied as a selective force for the removal of oligos that do not bind the target. At the end of the process, binding candidates are selected and tested separately. Importantly, previously described cell-specific aptamers have been reported to also have a secondary function subsequent to the binding. This has been usually achieved by an additional step in which the positive binders are tested again, this time at the cell level, for the secondary function[22–24]. This is done in a low throughput manner, separately testing each candidate.

The platform we describe here aims at achieving this goal and selects oligonucleotides with defined therapeutic functions, such as target cell apoptosis. The platform screens the oligonucleotide pool for the desired function, looking only for a chosen biological effect at the live cell level. A remarkable consequence of this feature is that the platform does not require any a priori biological knowledge about the target, such as which cell type it is, which surface markers it expresses, which mutations its genome or epigenome carry, and which current chemotherapies it is already resistant to. The platform only requires a clear identification of the input cells as the target or as a negative target.

This report describes the successful implementation of the platform on tumor cell lines as well as primary, patient-derived cells, resulting in the discovery of oligonucleotides capable of inducing selective and robust apoptotic response in these cells. The platform is found to be sufficiently robust to be highly reproducible in light of the variable nature of the targets. We describe the isolation of a particular, 54-nt oligonucleotide termed E8, which selectively induces apoptosis in patient-derived triple negative breast cancer cells. A polyethylene glycol (PEG)-modified version of E8 exhibited favorable biodistribution in animals, persistence in target tumors up to 48 hours after injection, and safety in human blood. Importantly, in patient-derived xenograft mice, E8 showed tumor reduction of approximately 50% within 10 days of treatment, and a robust effect in ex vivo organ cultures taken from chemotherapy-resistant triple negative breast cancer patients. With further improvement, these findings could lead to a drug discovery model which is target-tailored, mechanism-flexible, and nearly on-demand.

## Results

**Platform calibration.** The platform's workflow consists of two stages. The purpose of the first stage is to enrich a random initial single-stranded oligonucleotide library (~$10^{15}$) for specific target binders, however without notable reduction of its diversity. This is done by three rounds of a conventional cell-SELEX process (Supplementary Note 1). The enriched population then exits into the second, functional stage. The crucial challenge of this stage is that candidate functional oligonucleotides distribute evenly across the target cell sample, generating a very low effective concentration and are therefore highly unlikely to generate any noticeable response in cells. To overcome this issue and enhance the signal, we created an emulsion in which the library was dispersed together with microparticles, and performed emulsion PCR (ePCR) to coat each microparticle with multiple copies of a single oligonucleotide, or at most very few different ones. Thus each cluster creates a local high concentration for the oligonucleotide it holds (Supplementary Note 2). The process then starts and continues with the population of clustered oligonucleotides rather than a solution-phase library.

Each functional round commences by incubating the clustered library with target cells which were loaded with a reporter indicating a desired response, such as apoptosis or cell activation. Following incubation, the cell:cluster mixture is sorted by FACS to isolate cluster⁺/response⁺ events. Clusters are then eluted from cells, amplified by PCR, and amplified again by ePCR to generate new clusters for the next round. Two final analyses are then performed: first, all output libraries from the functional stage are experimentally compared in their ability to induce a response in the target cells; this test validates that the process has successfully driven the library towards improvement. Second, analysis by deep sequencing highlights the most successful oligonucleotides for further synthesis and functional validation.

**Platform verification.** This workflow was verified on a colorectal carcinoma cell line, HCT-116. HCT-116 cells first went through 7 rounds of cell-SELEX, followed by a binding assay (Fig. 1a, b). Library enrichment during the process was evaluated by sampling the first round, last round, and 2 selected intermediate rounds. Sampling was done mainly in order to save library material for additional studies. The output library from round 3 was then introduced into the functional stage for an additional 6–8 rounds. The functional stage is based on choosing a specific marker or

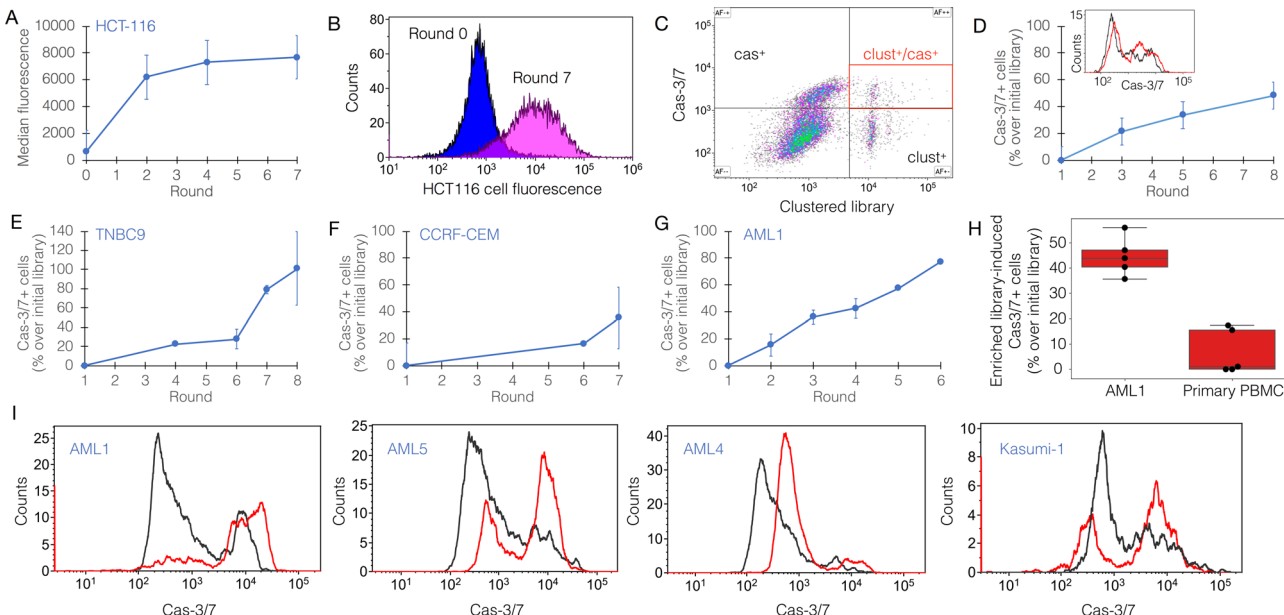

**Fig. 1 Tumoricidal oligonucleotide libraries created by response-directed in vitro evolution. a** Consistent enrichment of the binding capacity of a random library during the initial stage of the process, implemented on HCT116, a colorectal cancer cell line. Each point represents an independent binding assay (only rounds 0, 2, 4, 7 were sampled for enrichment analysis). **b** A representative binding assay showing the success of the initial stage (round 7 [final] vs round 0 of the process) on HCT116 cells. **c** Representative sorting plot during the functional stage of the process. Bead-clustered library (x-axis) is tagged by Cy5; Cas-3/7 is a green fluorescent reporter. Red gate within the upper right quadrant includes cas-3/7+ (cas+) cells bound to an oligo cluster (clust+). These events are sorted and carried forward to the next round. **d** Consistent enrichment of the ability to induce cas-3/7 activation in HCT116 cells by the oligo library. Inset shows flow cytometric analysis of cas-3/7 activity in HCT116 cells (black, cells treated with round 1; red, cells treated with round 8). **e–g** Representative runs of response-directed in vitro evolution, resulting in oligo libraries with cas-3/7 activity-inducing capacity (only rounds 1, 3, 5, 8 were sampled for enrichment analysis); **e** patient-derived xenograft (PDX)-derived triple negative breast cancer (TNBC) cells, termed TNBC9 (only rounds 1, 4, 6, 7, 8 were sampled); **f** human acute lymphoblastic leukemia cell line (CCRF-CEM; only rounds 1, 6, 7 were sampled, additional intermediate data points are missing due to insufficient material); **g** patient-derived acute myeloblastic leukemia (AML), termed AML1. **h** The selectivity of tumoricidal oligo library towards AML1 compared with primary peripheral blood mononuclear cells (PBMC) from a healthy donor. Shown is a representative analysis of cas-3/7 activity in both targets induced by the same library. The response observed in primary PBMC is statistically zero. **i** The exclusivity of a library evolved against AML1 target cells, to AML cells from other patients (AML4, AML5) and an AML cell line, kasumi-1. Shown is a flow cytometric analysis of cas-3/7 activity (black, round 1; red, round 6 [final] of the process).

mechanism for targeting by the library. As a functional reporter in this case study, we chose a fluorogenic substrate of activated caspase-3/7 (cas-3/7). This reporter produced a good signal for sorting. Sorting of cluster+/cas-3/7+ events went on for 8 rounds (Fig. 1c). In each round, the incubation time for generating response was 1.5 h. Importantly, cells entering the round being already dead are gated out based on their physical parameters, to prevent enrichment of dead cell-binding oligos, which are a potentially drastic contaminant. Strikingly, a comparison of the output libraries from all functional rounds demonstrated a consistent improvement in the library's ability to induce cas-3/7 activity in HCT-116 cells (Fig. 1d).

We repeated this workflow on several tumor targets of human origin: primary triple negative breast cancer (TNBC), an acute lymphoblastic leukemia (ALL) cell line, and primary acute myeloid leukemia (AML). TNBC cells (termed TNBC9) were produced from patient-derived xenografts as previously described[25]. MCF10A cells, a non-tumorigenic breast epithelial cell line[26], were used as negative target cells. These runs resulted in oligonucleotide libraries which exerted potent and selective cytotoxicity on the target cells, including those derived directly from patients (Fig. 1e, f, g, h). AML cells (termed AML1) were freshly isolated from patient's blood (Supplementary Note 3, Supplementary Figs. 1, 2), and peripheral blood mononuclear cells (PBMC) from a healthy donor were used as negative target cells. Here, too, the process successfully produced a library which induced cas-3/7 activation in AML cells but not in PBMC from a

healthy donor (Fig. 1g, h). To evaluate the exclusivity of libraries to the target cells used in their evolution, we examined the ability of the AML library described above (AML1) to induce apoptosis in other AML cells, both freshly-isolated from patients, and a cell line. Interestingly, some target cells exhibited partial resistance to this library, while others were susceptible (Fig. 1i).

**Lead selection and identification of E8.** In order to evaluate the applicability of this platform as a therapeutic strategy for cancer, we followed the TNBC9 targeting results with selection of lead molecules. Based on sequencing analysis (Fig. 2a), 10 candidate oligonucleotides were selected, synthesized, and folded. We did not observe significant homology ($p = 0.1$) within the sequence group or within the structures (Supplementary Table 1, Supplementary Note 4, Supplementary Figs. 3–5). The effectiveness of these candidates in target killing was measured on the PDX-derived TNBC9 cells, highlighting a single oligonucleotide, termed E8, as the most effective (Fig. 2b). A simulation and prediction of the 3D structure of E8 was performed (Fig. 2c, Supplementary Note 5, Supplementary Figs. 6–12). The observed level of direct target killing by E8 in vitro and ex vivo ranged between ~20–40% in independent biological replicate experiments, which is comparable to the levels observed with approved anti-cancer biologicals[27–29]. E8 demonstrated remarkable selectivity at the target cell level, killing TNBC9 but not MCF10A cells, which were used as negative targets in the in vitro evolution process (Fig. 2d). E8 was not exclusive to TNBC9 and showed a

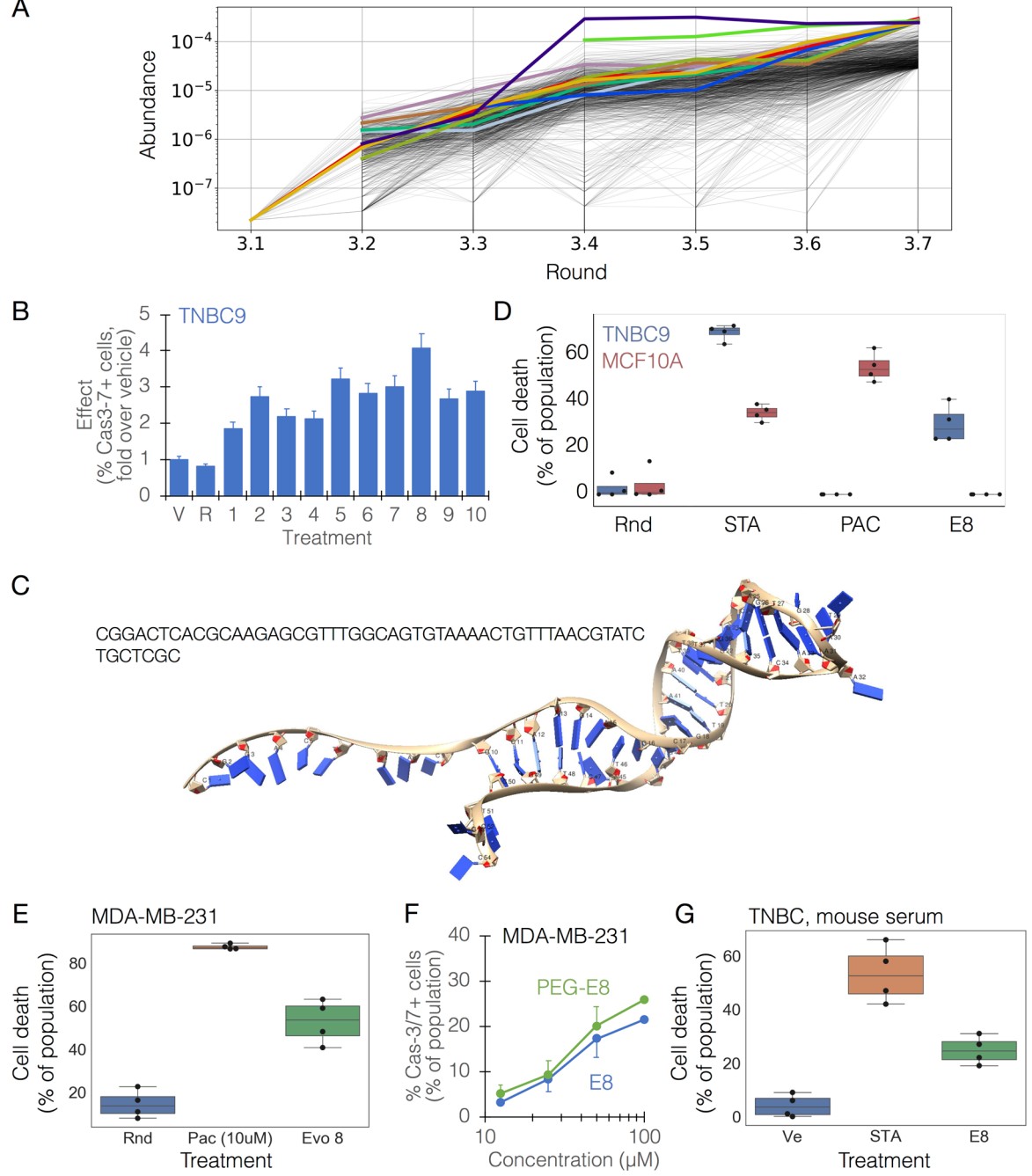

**Fig. 2 Identification of a lead candidate, E8, from a tumoricidal oligo library.** Cell types are denoted in the top left corner of plots. **a** Sequence abundance plot from a representative response-directed in vitro evolution run. The plot shows a random sample of 1,000 sequences out of the 10,000 most abundant sequences in each round (traces are shown from their first appearance in the data) with the 10 most abundant ones highlighted in color. These were synthesized and screened to find candidates. **b** A representative screening to highlight effective oligos. The response was measured as the ability to induce significant cas-3/7 activation in the population compared with vehicle. V, vehicle; R, random oligonucleotide; 1–10, oligo IDs (E1, E2, …, E10). **c** Simulated structure of E8 (see Supplementary Note 5 for derivation). **d** Selectivity of E8 to TNBC9 cells (blue) over the negative target cells, MCF10A (red). The response was measured by a cell viability count assay. STA, staurosporine; PAC, paclitaxel; Random, random oligonucleotide. **e** The effect of E8 (100 μM) on MDA-MB-231 cells. Rnd, random oligonucleotide; PAC, paclitaxel. **f** The dose-response curve for E8 (blue) and PEGylated (PEG)-E8 (green), showing persistence of effect in the modified oligo. The two curves are statistically indifferent. **g** The effect of E8 (100 μM) on TNBC cells in mouse serum. Ve, vehicle; STA, staurosporine. Chemotherapeutic controls were given at equivalent concentrations to E8.

remarkable effect on MDA-MB-231 cells as well (Fig. 2e). In preparation for in vivo testing, these effects were re-validated using E8 modified with polyethylene glycol (PEG), a modification that extends in vivo stability and half-life of the oligonucleotide, demonstrating that the effect was retained with PEG (Fig. 2f). In addition, E8 retained function in mouse serum (Fig. 2g).

**Biodistribution and safety of E8.** To determine the dispersion of E8 we used fluorescently-labeled E8 as previously described for aptamer in vivo imaging probes[30–32]. The molecule, modified with 5'-Cy5.5 and 3'-PEG, was injected intravenously in two doses (6 and 60 mg/kg) into NOD/SCID mice in which MDA-MB-231 tumors were induced. These experiments

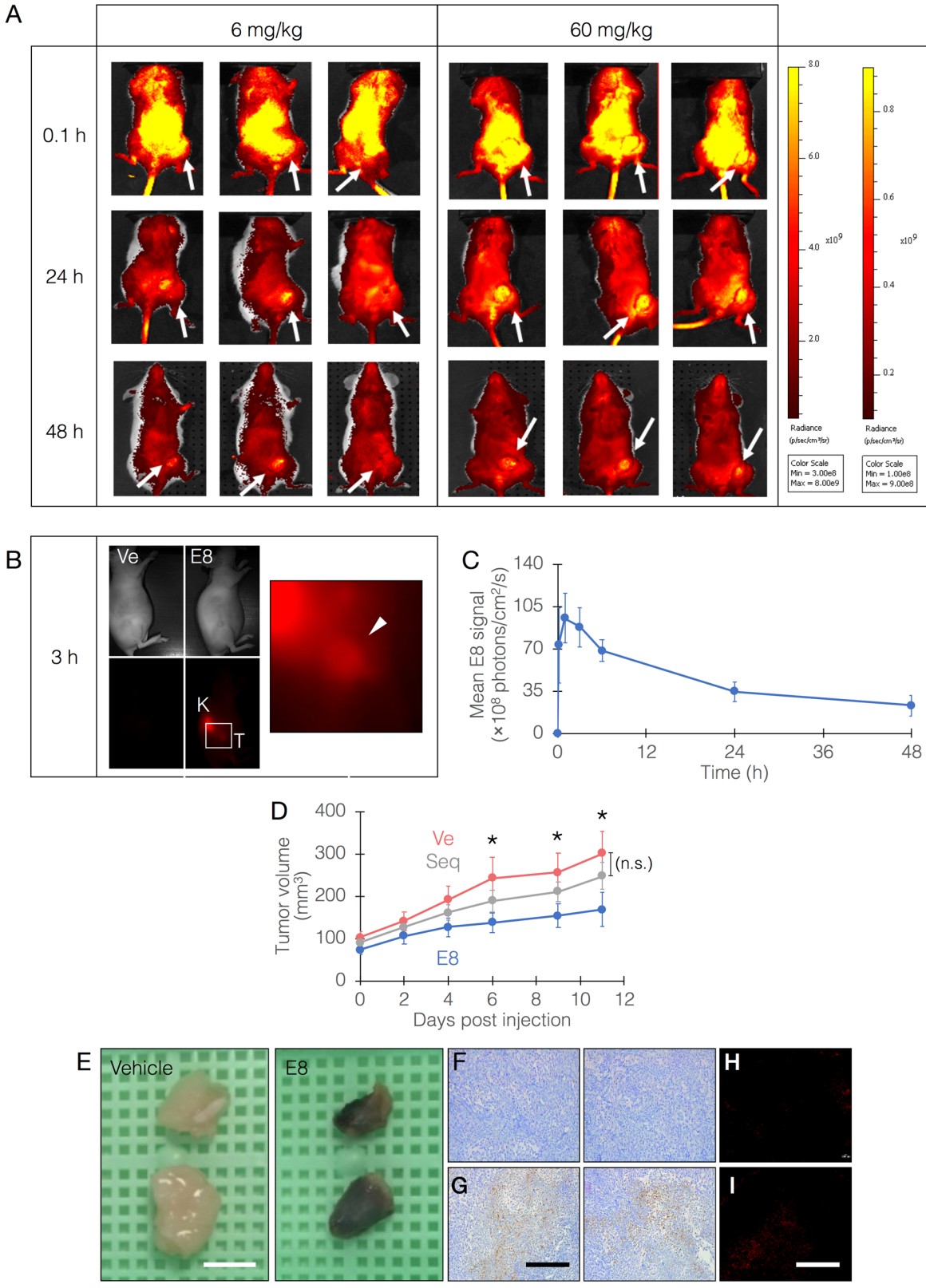

showed that E8 localizes to and is remarkably retained in the tumors at 24 and 48 h post injection (Fig. 3a, b, c). Fluorescently-labeled control oligos have consistently shown no accumulation in the tumors, and were localized to the liver and kidneys as rapidly as 30 min (Supplementary Note 6, Supplementary Figs. 13–14), in agreement with previously reported studies[30,31]. Multiple experiments with random

sequence controls revealed no significant protease resistance of E8 ($p = 0.08$) which could be attributed to its structure. We therefore concluded that this persistence is most likely due to PEGylation of E8. Furthermore, when E8 was mixed with whole human blood from healthy donors, no hemolysis, agglutination, or cytokine responses were observed (Supplementary Note 7, Supplementary Figs. 15–17).

**Fig. 3 E8 biodistribution and efficacy in an animal model. a** E8, modified with Cy5.5 and PEG, was injected at 6 or 60 mg/kg, i.v. into NOD/SCID mice bearing MDA-MB-231-derived tumors on their right hind limb. Fluorescence was measured in vivo immediately after injection and following 24 and 48 h. White arrows point to tumor locations. Color scales are indicated on the right; left scale relates to 6 mg/kg, right scale to 60 mg/kg. **b** Homing and retention of E8 at tumor site 3 h following i.v. injection (Ve, vehicle; K, kidney). Inset region is shown magnified on the right. White arrowhead points to tumor site. **c** Quantitative measurement of E8 level in tumors up to 48 h post injection. E8 level peaks at 1–3 h post injection, then fall but is still maintained later. **d** The efficacy of E8 in mice bearing MDA-MB-231-derived tumors. E8 was injected at 100 mg/kg, 1 dose/2 d for 11 d, and tumor volumes were measured. Mean tumor volumes of the two groups are statistically indifferent at day 0. Asterisks denote a statistically significant difference with $p < 0.05$ ($n = 8$ mice/group). Ve, vehicle. Seq, control oligo. **e** Representative photographs of tumors excised from mice sacrificed at day 11. Tumors from E8-treated mice appear necrotic. Scale bars, 5 mm. **f, g** histochemical analysis of caspase-3 activity in tumor-derived tissue sections (**f**, vehicle-treated; **g**, E8-treated). Caspase-3 activity is exhibited as brown color. Scale bar, 500 μM. **h, i** TUNEL analysis of tumor-derived tissue sections (**h**, vehicle-treated; **i**, E8-treated). See Supplementary Fig. 18 for a magnified version of these panels. Scale bar, 500 μM.

**Efficacy of E8 in vivo**. To evaluate the efficacy of E8, the PEGylated oligonucleotide was injected once/2 days during the course of an 11-day period, at a dose of 100 mg/kg (equivalent in molar terms to standard chemotherapy). During this period, in E8-treated animals, tumor growth was inhibited, with mean tumor volumes significantly lower ($p = 0.04$) than in vehicle-treated animals (final volumes: $168 \pm 39$ vs $301 \pm 51$ mm$^3$ in E8-treated animals and vehicle-treated ones, respectively) (Fig. 3d). Treatment with a control oligo showed no significant effect on tumor size ($p = 0.08$). Remarkably, tumors extracted from E8-treated animals exhibited macroscopic signs of tissue death (Fig. 3e). Analysis of caspase-3 activity in histological sections from these tumors showed remarkable staining in tumors from E8-treated animals (Fig. 3f, g), reinforcing the hypothesis that this effect was caused directly by E8, which was selected from a library evolved specifically to activate caspase-3. Tissue sections were also analyzed by TUNEL showing marked effect in E8-treated tumors (Fig. 3h, i; Supplementary Note 8, Supplementary Fig. 18). Importantly, no changes in appearance or body weight were observed following injections (Supplementary Note 6).

**Efficacy of E8 in ex vivo organ cultures**. The efficacy of E8 was also evaluated in human ex vivo organ cultures (EVOC)[7,33] freshly derived from BC patients (Supplementary Note 9, Supplementary Fig. 19). The pathological assessment, performed in a blinded manner by two experts, showed that E8 had a significant effect (grades 3–4 on a 0–4 scale) on tumor cells in the EVOC samples from 2 patients, both of them showing resistance to at least one chemotherapy (Fig. 4a, b).

The described platform was reproducibly tested in $n = 9$ independent runs on human tumor targets from different types and sources (this report describes runs on 7 different targets, out of which 2 targets were run twice), with each library tested in multiple biological repeats. It is interesting to note that the platform successfully produced effective libraries against targets with known resistance to multiple drugs, suggesting that the process is driven sufficiently robustly so as to find solutions to targets following significant biological alterations (e.g., shutting down pathways to resist a drug).

## Discussion

We describe a platform for the relatively rapid de novo discovery of therapeutic oligonucleotides by response-directed in vitro evolution. This platform could potentially address the central limitations of our current model of drug discovery. Particularly, this platform receives a human sample and operates a specific algorithm to generate a new therapeutic molecule tailored to the sample. The current algorithm can be improved based on our findings. For example, these findings indicate that output libraries and candidate oligos are not absolutely exclusive to the target cells

used as input in their evolution process. Therefore, the personalized algorithm should include an early step that screens any incoming sample against the library of previously-generated oligonucleotides, to shunt directly to synthesis in case effective and selective candidates are found. The algorithm is still personalized per sample, but such decision trees could tremendously improve its efficiency. We are also currently improving methods for candidate selection from sequencing data, based on parameters orthogonal to abundance.

A crucial factor in the success of target-tailored therapy is discovery speed. In developing the described approach, an emphasis has been put in confining the overall discovery time to several days, which is within the time scale of approved personalized therapies such as CAR-T cells. It is important to note, that some recent methods for the selection of aptamers are significantly faster than this: several reported methods require only a single round of selection to identify monovalent and bivalent aptamers[34–36]. Additional efficient modalities for aptamer selection include capillary electrophoresis[37–39] and magnetic sorting[40]. Electrophoresis and magnetic beads have been combined with microfluidics to create integrated systems for aptamer selection[41–43]. All these methods have demonstrated that aptamer selection could be significantly sped up and multiplexed. Moreover, recent works have explored new ways for enhancing functional selection capabilities, with remarkable results. For example, an in vivo selection method has successfully identified aptamers capable of crossing the blood–brain barrier[44]. A method termed pheno-SELEX has been developed to select aptamers specifically against the invasive phenotype of tumor cells[45]. We note that these and other works, also reviewed elsewhere[46], highlight the potential of selecting functional aptamers rapidly and efficiently, and could significantly improve our approach for discovering therapeutic oligonucleotides.

Although the test case was cancer, our findings highlight the possibility to utilize this platform against other targets, such as antibiotic-resistant bacteria, which are a challenge of increasingly critical importance. Central to this challenge is the profound asymmetry between the period of time required for the development of a new antibiotic and the period of time required for targets to develop resistance to it: while new drugs require on average more than 10 years and $2.6B to develop[47], antibiotic resistance in bacteria could arise within a few generations, or on the scale of hours[48,49], rendering the pharmaceutical industry extremely limited in dealing with this challenge. The present platform could offer a significant advantage in this battle. Interestingly, oligonucleotides have the additional unique advantage of being "digitizable"; they can be distributed as electronic sequence files and synthesized locally, owing to their facile synthesis. We have recently shown using network models that this concept formulates the most effective strategy to date to mitigate global pandemics[50]. Coupled with an ultra-rapid discovery system, a tool is created which, arguably, must be pursued.

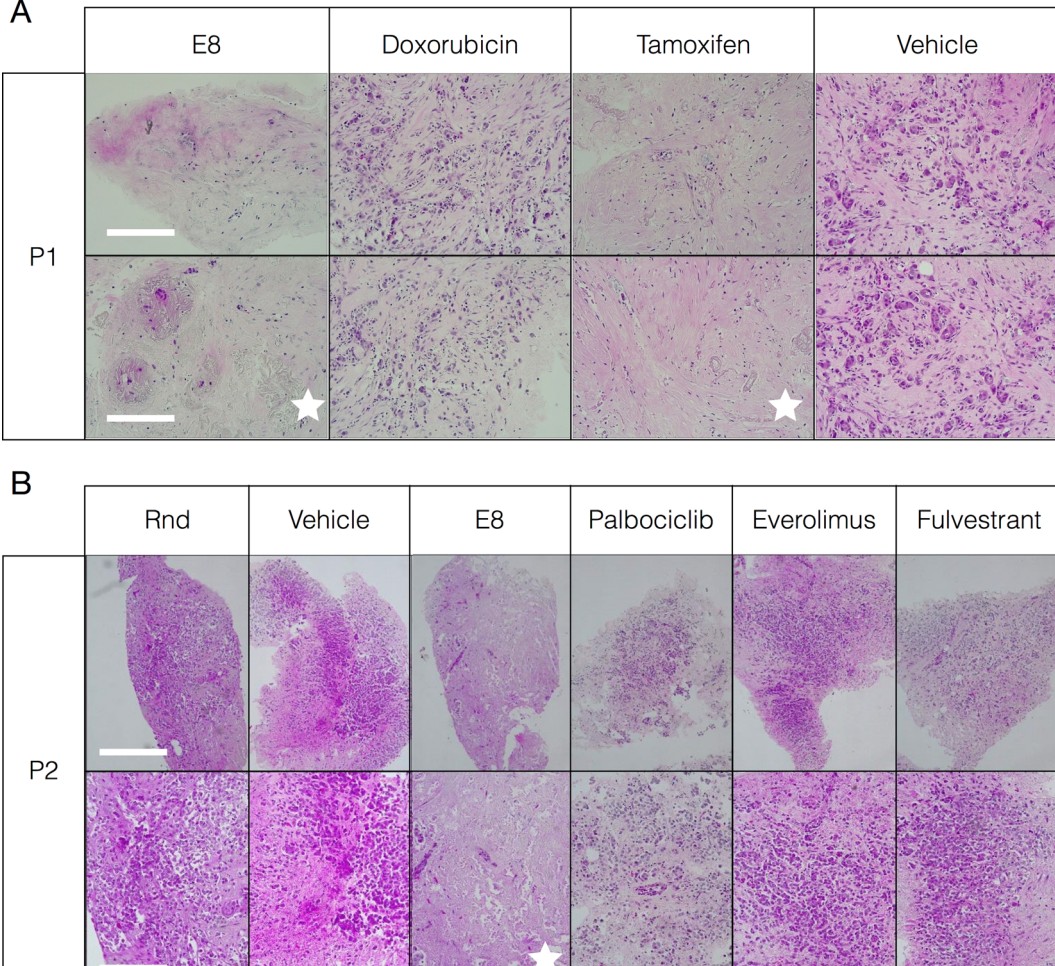

**Fig. 4 Efficacy of E8 in human ex vivo organ cultures (EVOC).** E8 was administered to EVOCs freshly derived from TNBC patients (two representative patients, P1 and P2, are shown). **a, b** E8 and other chemotherapies were administered. E8 was administered at 50 µM, while the chemotherapeutic drugs were administered at their respective therapeutic doses. Treatment was given 1/d for 2 d, and samples were fixed at 5 d, sections were made and stained with hematoxylin-eosin. Effects were graded by 2 blinded pathologists on a 0–4 scale. White stars denote the experimental groups in which effect reached a grade of at least 3, and relate to the entire respective column. Rnd, random oligonucleotide; magnifications are shown in the bottom left corners of each row. Scale bars: A, upper and lower scale bar, 200 µm. B, upper scale bar, 1 mm; lower scale bar, 200 µM.

## Methods

**Cell lines and primary cells**. Cell lines were purchased from American Type Cell Culture (ATCC). Human primary acute myeloblastic leukemia (AML) cells were isolated from donors by standard procedures (Institutional Review Board [IRB] approval numbers 0297–15-TLV & 4573–17-SMC). Human primary PBMCs were isolated from healthy donors by standard procedures (IRB approval number 0297–15-TLV). Human (PDX-derived) triple negative breast cancer cells were a kind.pngt from B. Dekel, Sheba Medical Center, Israel. Target blasts were isolated by magnetic sorting using a commercial kit (Miltenyi Biotec) according to the manufacturer's instructions.

**DNA libraries and cell-SELEX**. DNA libraries were designed with a 50-nt random core flanked by 20-nt constant regions and ordered from Integrated DNA Technologies (IDT, 5 umol scale). Randomization was done by hand mixing at IDT. All libraries passed in-house QC of uniformity by HPLC prior to usage. Libraries were reconstituted in ultrapure water at a stock concentration of 1 mM. Cell-SELEX was performed as follows. An ssDNA library constructed of a random core flanked by constant regions is folded in the presence of constant region-complementary oligonucleotides (termed caps). Folding was carried out by incubation at 95 °C for 5 min, cooling on ice for 10 min, and an additional 10 min incubation at 37 °C. Folded library and cells were incubated together in the target cell medium supplemented with 10% human serum for 1 h. Library concentration in the incubation step was set to 500 nM. After each round, the sample was washed to dilute unbound candidates $10^4$-fold for the first selection round and $10^6$-fold from the second round forth. To prepare the next round's input library, the bound fraction was eluted by incubation at 95 °C for 10 min. From the 2nd round on a negative selection was added. The eluted library was folded again and incubated with the

non-target cells as described above, this time the unbound fraction is taken as an input for an asymmetric PCR (aPCR) process. ssDNA was purified from the aPCR product using preparative HPLC on an Agilent 1100 instrument. Samples of output libraries from all rounds were stored for evaluation.

**Functional in vitro evolution**. Clustered libraries on Ion Proton spheres were generated using Ion Proton sample prep Ion PI Hi-Q OT2 200 Kit and an Ion OneTouch automated sample prep system. The Ion PI Hi-Q OT2 200 Kit user manual instructions were followed. Enrichment QC was done using the Ion Sphere Quality Control Kit according to the manufacturer instructions. Ion spheres were labeled using Cy5 conjugated caps in order to help with their detection in the melody FACS.

**DNA sequencing**. Sequencing was done on an Illumina NextSeq 500 sequencer using NextSeq 500/550 High Output Kit according to the manufacturer's instructions.

**Fluorescence-activated cell sorting and microscopy**. Sorting was performed on a Becton-Dickinson FACSMelody cell sorter equipped with blue, red, and violet lasers (9 color configuration). Cell imaging was done on a Nikon Eclipse Ti2 fluorescent microscope with a Chroma-49004 or Chroma-49006 filter cubes, Lumencor Sola SE II 365 illumination, and an integral $CO_2$ incubator. Scans were analyzed using NIS Elements AR_software. Cells were identified by Hoechst staining and apoptosis was determined upon co-location with CellEvent™ Caspase-3/7 Green Detection Reagent labeling (see Supplementary Note 10 for additional markers evaluated).

**Flow cytometry**. Flow cytometry was performed on a Becton-Dickinson Accuri C6 Plus cytometer equipped with 488 nm and 630 nm lasers, and on a Beckman-Coulter Cytoflex cytometer with a B5-R3-V5 laser configuration.

**Oligonucleotide synthesis**. Synthesis of selected oligonucleotides including any modification, for both validation and large scale (>1 mmol) experiments, were done by IDT and LGC Axolabs.

**In vivo experiments**. All animal procedures were performed in the facilities of Science in Action Ltd. (Rehovot, Israel) (National Ethical Approval number 17–3–113). Animals used in this study were female nude mice 9–10 weeks old. Mice (a total of 8 mice/group) were subcutaneously injected (clipping at approximately 24 h prior to injection) with MDA-MB-231 cells ($3 \times 10^6$ cells/mouse) into the right flank, 0.2 mL/mouse (using tuberculin syringe with 27 G needle). E8 was injected S.C. once/2 days at 100 mg/kg dose for a period of 11 days. The administration was performed at a constant volume dosage based on individual body weights using a 1 mL insulin syringe with 30 G needle.

**Ex vivo organ cultures**. Ex vivo organ cultures (EVOC) were prepared by Cureesponse Ltd. (IRB approval number 0656–18-TLV), stained with H&E, and analyzed by 2 blinded pathologists. Statistical analysis was performed by student's $t$-test assuming equal variances. Structure prediction of E8 was performed by a Profacgen Inc. as described in Supplementary note 5.

**Statistics and reproducibility**. The described platform was reproducibly tested in $n = 9$ independent biological repeats on human tumor targets from different types and sources. All experimental datasets include groups with at least $n = 3$ samples. Confidence and $p$-values were calculated using equal-variance $t$-test. Flow cytometric datasets include at least $n = 1000$ events. Sequencing data was analyzed statistically by standard methods as previously described.

**Reporting Summary**. Further information on research design is available in the Nature Research Reporting Summary linked to this article.

## Data availability

All data and software used in this work will be made available upon request to the authors.

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

## Acknowledgements
The authors wish to thank Dr. Benjamin Dekel (Sheba Medical Center, Ramat Gan, Israel) for the kind gift of TNBC9 cells; to Dr. Seth Salpeter, Dr. Vered Bar, and Ms. Sarah Baum (Curesponse Ltd, Tel Aviv, Israel) for EVOC experiments; to Dr. Anat Globerson-Levin for technical assistance with in vivo biodistribution experiments; and to the entire team at Augmanity and Aummune Ltd. for valuable technical assistance and discussions.

## Author contributions
The following authors designed experiments, performed experiments, analyzed data, and wrote the manuscript: N.M., Y.A., E.L., I.R and I.B. The following authors designed experiments, performed experiments, and analyzed data: N.S., L.B., G.H. and G.L. The following authors performed experiments and analyzed data: E.D., M.G., A.Z., A.P. and N.R. I.A. and A.S. designed experiments and provided valuable materials. A.A.H. and G.N. provided valuable technical assistance and oversight support. I.B. oversaw the project.

## Competing interests
The authors declare no competing non-financial interests, but the following competing financial interests: all authors are employees and shareholders in companies that develop technologies described in this article (N.M., L.B., G.H., I.R., A.Z., G.L., A.A.H., I.B. at Augmanity; E.L., N.S., E.D., A.P., Y.A., G.N. at Aummune). The following authors are listed as inventors on patent applications related to technologies described in this article: I.B., N.M., I.R., G.H., Y.A., E.L., A.A.H. (PCT/IB18/00418, pending); I.B., N.M., E.L., L.B., E.D., I.R., G.H., Y.A., A.A.H. (PCT/IB18/00613, pending); I.B., N.M., A.A.H., Y.A., L.B., E.D., E.L., I.R., N.S. (62/738,235, pending).
