## [Peer Review File · Communications Biology]

Reviewers' comments:

Reviewer #1 (Remarks to the Author):

In this manuscript, the authors present an approach to develop therapeutic anti-tumor aptamers which has the potential to be individualized, rapid and economically repeatable. This is a highly original paper which is well written and logically argued. The authors present a workflow of standard cell-SELEX for 7 rounds followed by functional screening, in this case, of caspase 3/7 activity across several different cell lines including patient-derived lines. They show effectiveness to activate caspase in cell lines, show fluorescently labelled aptamers in mice efficacy, and also shown in human organ cultures. This is rather impressive to go all the way from aptamer selections to organ culture studies.

1. The authors should better explain why caspase 3/7 was chosen as the functional screen. Were other functional screens tested and compared? If so that data should be shown.
2. Please provide the ethical approval for the human samples (eg AML cells from patient blood and PBMC from healthy donor). I can only see the animal ethical approval.
3. How much sequence homology is there amongst the oligos in Figure 2? E1-E10? Where are the aptamer sequences?
4. What is dose response data for 2D and 2F? Also for paclitaxel/staurosporine. How about PEGylation for the assay in 2D?
5. Quite remarkable that the E8 persists for 48 hours in Figure 3. How about random control sequences – is persistence due to a nuclease resistant structure of the aptamer or all just due to PEG?
6. Fig 3 needs scrambled controls to show specificity rather than general effect of small nucleic acids getting localized to tumor sites.
7. Fig 3 H/I TUNEL data cannot see anything.
8. Fig 3 F/G why are there two panels?
9. Fig 3 all panels need scale bars.
10. Fig 4 needs scale bars.
11. Fig 4 20-50 μ M? Which panel?

Reviewer #2 (Remarks to the Author):

In "Discovery of tumoricidal DNA oligonucleotides by effect-directed in-vitro evolution", Mamet et al describe an aptamer selection method that uses minimal binding selection rounds followed by selection rounds based on the cellular response, in this case cas-3/7, in vitro. After the initial selection rounds they use ePCR on microparticles to create many copies of the DNA on a particle. This is similar to previous bead-based methods, although the current method has a much greater diversity of sequences. Using a variety of cell lines, the authors show a reduction in tumor size without effecting non-cancerous cells. The manuscript is well-written but comparison to other bead-based methods should be discussed.

1. I did not find a way to access the many Supplementary Notes. I need to read those.
2. I would recommend calling this method "response-directed" because you are measuring the cells' response (cas-3/7) to the aptamers (ODNs), rather than the effect (cell killing) of the ODN. Previous methods like In vivo SELEX (Whole Animal SELEX) also measured an effect (Binding in vivo; cell reduction).
3. The method does not seem to be any more rapid than many other methods which now only use 1-2

rounds of selection.

4. No description was given of what the sequence space looks like for a given reduced library. Is E8 one of thousands of similar sequences in these data?

5. Other methods use 1 or 2 rounds of selection, followed immediately by FACS selection in a manner similar to yours.

6. The unique parts of this method are the few SELEX rounds followed by ePCR to create an improved bead-based (microparticle) method, and the subsequent cell-response detection (in the case of cas-3/7, but not in the case of death) for sequence selection.

7. More discussion of other methods (bead-based, whole-animal selex, etc) should be inserted into this manuscript.